# Active Exploration for
# Inverse Reinforcement Learning

**David Lindner**
Department of Computer Science
ETH Zurich
david.lindner@inf.ethz.ch

**Andreas Krause**
Department of Computer Science
ETH Zurich
krausea@ethz.ch

**Giorgia Ramponi**
ETH AI Center
giorgia.ramponi@ai.ethz.ch

## Abstract

Inverse Reinforcement Learning (IRL) is a powerful paradigm for inferring a reward function from expert demonstrations. Many IRL algorithms require a known transition model and sometimes even a known expert policy, or they at least require access to a generative model. However, these assumptions are too strong for many real-world applications, where the environment can be accessed only through sequential interaction. We propose a novel IRL algorithm: **Ac**tive **e**xploration for **I**nverse **R**einforcement **L**earning (AceIRL), which actively explores an unknown environment and expert policy to quickly learn the expert's reward function and identify a good policy. AceIRL uses previous observations to construct confidence intervals that capture plausible reward functions and find exploration policies that focus on the most informative regions of the environment. AceIRL is the first approach to active IRL with sample-complexity bounds that does not require a generative model of the environment. AceIRL matches the sample complexity of active IRL with a generative model in the worst case. Additionally, we establish a problem-dependent bound that relates the sample complexity of AceIRL to the suboptimality gap of a given IRL problem. We empirically evaluate AceIRL in simulations and find that it significantly outperforms more naive exploration strategies.

## 1 Introduction

Reinforcement Learning (RL; Sutton and Barto, 2018) has achieved impressive results, from playing video games (Mnih et al., 2015) to solving robotic control problems (Haarnoja et al., 2019). However, in many applications, it is challenging to design a reward function that robustly describes the desired task (Amodei et al., 2016; Hendrycks et al., 2021). Instead of using an explicit reward function, Inverse Reinforcement Learning (IRL; Ng et al., 2000) seeks to recover the reward by observing an *expert*, e.g., an human who already knows how to perform a task. However, most existing IRL algorithms assume that the transition model, and in some cases, the expert's policy, are *known*. In many real-world applications, this is not given, and the agent needs to estimate the transition dynamics and the expert policy from samples. Figure 1 shows an illustrative example where the agent can choose between different paths that have different properties, e.g., walking speeds, and lead to different goals. The agent has to explore the environment and query the expert policy in order to infer the expert's reward function.

IRL with sample-based estimation was only recently analyzed formally by Metelli et al. (2021). They decompose the error on the reward into a contribution from estimating the transition model and

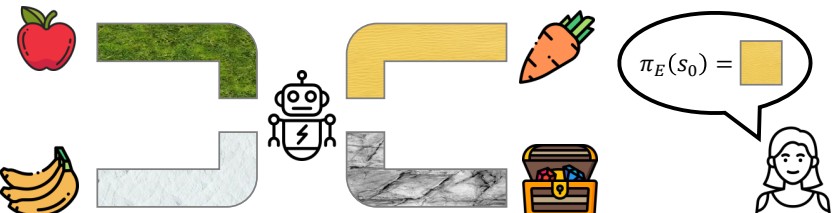

Figure 1: An illustrative example of Active IRL. The agent can choose between four paths that lead to different objects. It can get action recommendations from an expert but does not know about the properties of the different paths (the transition dynamics) or the value of different items (the reward function). The agent's goal is to infer which reward functions explain the expert's recommendations. Only observing the expert actions is not enough to do that. Rather, the agent has to explore the environment and learn about the dynamics. The human might prefer to find the treasure over the carrot but still recommend the yellow path because the treasure is very difficult to reach. To explore efficiently, the agent has to combine its uncertainty about the expert policy with its uncertainty about the environment to choose where to explore. AceIRL implements an exploration strategy that aims to infer which reward functions are consistent with the expert's recommendations as quickly as possible. We present experiments on a version of this environment in Section 7.

estimating the expert policy. Based on this, Metelli et al. (2021) propose an efficient sampling strategy to recover a good reward function. However, they assume a *generative model* of the environment, i.e., the agent can query the transition dynamics for arbitrary states and actions. In practice, this assumption is unrealistic. The agent in Figure 1 starts in the middle and cannot learn about the properties of any path without actually *exploring the environment*.

In this work, we consider IRL with unknown transition dynamics and expert policy and focus on exploring the environment in order to recover the expert's reward function efficiently. To the best of our knowledge, we present the first paper providing sample complexity guarantees for the active IRL problem without access to a generative model.

Our main contributions are:

- We propose the active IRL problem in a finite-horizon, undiscounted Markov Decision Process (MDP) and characterize necessary and sufficient conditions for solving it (Section 4.1).

- We analyze how the estimation errors of the transition model and the expert policy contribute to the estimation error of the reward function, extending prior work to the finite-horizon setting. We provide a novel analysis of how this error affects the performance of the policy, which optimizes the recovered reward function (Section 4.2).

- We propose a novel algorithm, **Ac**tive **e**xploration for **I**nverse **R**einforcement **L**earning (AceIRL), which actively explores the environment and the expert policy to infer a good reward function. In each iteration, AceIRL constructs an exploratory policy based on the estimation error of the recovered reward function (Section 6).

- We consider two different exploration strategies for AceIRL. The first, more straightforward, strategy provides a sample complexity similar to the algorithm proposed by Metelli et al. (2021), which has access to a generative model (Section 6.1). The second strategy takes the expected reduction in uncertainty into account (Section 6.2). This yields a tighter, problem-dependent sample complexity bound at the cost of solving a convex optimization problem in each iteration (Section 6.3).

- We evaluate AceIRL empirically in simulated environments and demonstrate that it achieves significantly better performance than more naive exploration strategies (Section 7). We provide code to reproduce out experiments at `https://github.com/lasgroup/aceirl`.

The proofs of all results presented in the main paper can be found in Appendix B.

## 2 Related Work

Most IRL algorithms assume that the underlying transition model is known (Ratliff et al., 2006; Ziebart et al., 2008; Ramachandran and Amir, 2007; Levine et al., 2011). However, the transition model usually needs to be estimated from samples, which induces an error in the recovered reward function that most papers do not study. Metelli et al. (2021) analyze this error and the sample complexity of IRL in a tabular setting with a generative model. They propose an algorithm focused on transferring the learned reward function to a fully known target environment. Dexter et al. (2021) provides a similar analysis in continuous state spaces and discrete action spaces, but they still require a generative model of the environment. In contrast, we *do not* assume access to a generative model and thus need to tackle the exploration problem in IRL.

Some prior work studies active learning algorithms for IRL in a Bayesian framework but without theoretical guarantees. Lopes et al. (2009) propose an active learning algorithm for IRL that estimates a posterior distribution over reward functions from demonstrations, requiring a prior distribution and full knowledge of the environment dynamics. Relatedly, Cohn et al. (2011) consider a Bayesian IRL setting with a semi-autonomous agent that asks an expert for advice if it is uncertain about the reward. Brown et al. (2018) empirically study active IRL in several safety-critical environments, selecting queries using value at risk. Kulick et al. (2013) consider active learning for a robotic manipulation task, asking a human expert for advice in situations with the highest predictive uncertainty. Similarly, Losey and O'Malley (2018) propose a method to learn uncertainty estimates from human corrections in a robotics context. All of these papers assume a Bayesian framework and do not provide theoretical guarantees. In contrast, our setup does not require a prior over reward functions, and we provide theoretical sample complexity guarantees for our algorithm.

A separate line of work studies sample complexity in *imitation learning* where the goal is to imitate an expert policy rather than infer a reward function (Rajaraman et al., 2020; Xu et al., 2020). In particular, Abbeel and Ng (2005) also focus on exploration and propose to use the expert policy to explore relevant regions, whereas Shani et al. (2022) use an upper-confidence approach to exploration. Our setting is different because we focus on IRL instead of imitation learning, and we aim to explore to infer a reward function learn most effectively.

## 3 Preliminaries

Let us first introduce necessary background and notation that we use throughout the paper.

**Markov decision process.** A finite-horizon (or episodic) Markov Decision Process without reward function (MDP\R) is a tuple $\mathcal{M} := (\mathcal{S}, \mathcal{A}, P, H, s_0)$, where $\mathcal{S}$ is the finite state space of size $S$; $\mathcal{A}$ is the finite action space of size $A$; $P : \mathcal{S} \times \mathcal{A} \to \Delta_{\mathcal{S}}$ is the transition model; $H$ is the horizon and $s_0$ is the initial state. [1] In other words, a finite-horizon MDP\R is a finite-horizon MDP (Puterman, 2014) without the reward function. We describe an agent's behaviour with a (possible stochastic) policy $\pi \in \mathcal{S} \times [H] \to \Delta_{\mathcal{A}}$.

**Reward function.** A reward function $r : \mathcal{S} \times \mathcal{A} \times [H] \to [0, R_{\max}]$ maps state-action-time step triplets to a reward. Given an MDP\R $\mathcal{M}$ and a reward function $r$, we denote the resulting MDP by $\mathcal{M} \cup r$.

**Value functions and optimality conditions.** We define the *Q-function* $Q_{\mathcal{M} \cup r}^{\pi,h}(s, a)$ and *value-function* $V_{\mathcal{M} \cup r}^{\pi,h}(s)$ of a policy $\pi$ in the MDP $\mathcal{M} \cup r$ at time step $h$, state $s$ and action $a$ as:

$$Q_{\mathcal{M} \cup r}^{\pi,h}(s, a) = r_h(s, a) + \sum_{s',a'} \pi_{h+1}(a'|s') P(s'|s, a) Q_{\mathcal{M} \cup r}^{\pi,h+1}(s', a'); \ V_{\mathcal{M} \cup r}^{\pi,h}(s) = \sum_a \pi_h(a|s) Q_{\mathcal{M} \cup r}^{\pi,h}(s, a)$$

We define the *advantage function* $A_{\mathcal{M} \cup r}^{\pi,h}(s, a)$ as $A_{\mathcal{M} \cup r}^{\pi,h}(s, a) = Q_{\mathcal{M} \cup r}^{\pi,h}(s, a) - V_{\mathcal{M} \cup r}^{\pi,h}(s)$. A policy $\pi$ is optimal if $A_{\mathcal{M} \cup r}^{\pi,h}(s, a)$ is $\leq 0$ for each time step $h \in [H]$, state $s \in \mathcal{S}$, action $a \in \mathcal{A}$. We denote the set of optimal policies for the MDP $\mathcal{M} \cup r$ with $\Pi_{\mathcal{M} \cup r}^*$.

---

[1] We can model any initial state distribution as a single initial state by modifying the transitions.

**State-visitation frequencies.** We define $\eta_{\mathcal{M},\pi}^{h,h'}(s|s_0)$ as the probability of being in state $s$ at time $h' \geq h$ following policy $\pi$ in MDP\R $\mathcal{M}$ starting in state $s_0$ at time $h$. We can compute it recursively:

$$\eta_{\mathcal{M},\pi}^{h,h}(s'|s) := \mathbb{1}_{\{s'=s\}} \quad \text{and} \quad \eta_{\mathcal{M},\pi}^{h,h'+1}(s'|s) := \sum_{s'',\tilde{a}} P(s'|s'',\tilde{a})\pi_{h'}(\tilde{a}|s'')\eta_{\mathcal{M},\pi}^{h,h'}(s''|s).$$

We can define the visitation frequencies for state-action pairs analogously (see Appendix B.1).

## 4 Active Learning for Inverse Reinforcement Learning (Active IRL)

In this section, we first introduce the Active Inverse Reinforcement Learning problem with and without a generative model (Section 4.1). Then, we define the feasible reward set for finite-horizon MDPs (Section 4.2) and characterize the error propagation on the reward function and the value function (Section 4.3), extending results by Metelli et al. (2021) to the finite horizon setting.

### 4.1 Problem Definition

Our goal is to design an exploration strategy to construct a dataset of demonstrations $\mathcal{D}$ such that an arbitrary IRL algorithm can recover a *good* reward function from it. To be agnostic to the choice of IRL algorithm, we consider the set of all feasible reward functions for a specific expert policy. Formally, we consider IRL problems $(\mathcal{M}, \pi^E)$ consisting of an MDP\R and an expert policy $\pi^E$, and we define the feasible reward set as follows.

**Definition 1** (Feasible Reward Set). *A reward function $r$ is feasible for an IRL problem $(\mathcal{M}, \pi^E)$, if and only if the expert policy $\pi^E$ is optimal in $\mathcal{M} \cup r$. We call the set of all feasible reward functions $\mathcal{R}_{\mathcal{M} \cup \pi^E}$ the feasible reward set. If we estimate the transition model and expert policy from samples, we refer to the recovered feasible set $\mathcal{R}_{\hat{\mathfrak{B}}} = \mathcal{R}_{\widehat{\mathcal{M}} \cup \hat{\pi}^E}$ in contrast to the exact feasible set $\mathcal{R}_{\mathfrak{B}} = \mathcal{R}_{\mathcal{M} \cup \pi^E}$.*

Now, we can formalize the goal of Active IRL as finding a exploration strategy that satisfies the following PAC optimality criterion.

**Definition 2** (Optimality Criterion). *Let $\mathcal{R}_{\mathfrak{B}}$ be the exact feasible set and $\mathcal{R}_{\hat{\mathfrak{B}}}$ be the feasible set recovered after observing $n \geq 0$ samples collected from $\mathcal{M}$ and $\pi^E$. We say that an algorithm for Active IRL is $(\epsilon, \delta, n)$-correct if after $n$ iterations with probability at least $1 - \delta$ it holds that:*

$$\inf_{\hat{r} \in \mathcal{R}_{\hat{\mathfrak{B}}}} \sup_{\hat{\pi}^* \in \Pi^*_{\widehat{\mathcal{M}} \cup \hat{r}}} \max_{s,a,h} \left| Q_{\mathcal{M} \cup r}^{\pi^*,h}(s,a) - Q_{\mathcal{M} \cup r}^{\hat{\pi}^*,h}(s,a) \right| \leq \epsilon \quad \textit{for each } r \in \mathcal{R}_{\mathfrak{B}},$$

$$\inf_{r \in \mathcal{R}_{\mathfrak{B}}} \sup_{\pi^* \in \Pi^*_{\mathcal{M} \cup r}} \max_{s,a,h} \left| Q_{\mathcal{M} \cup r}^{\pi^*,h}(s,a) - Q_{\mathcal{M} \cup r}^{\hat{\pi}^*,h}(s,a) \right| \leq \epsilon \quad \textit{for each } \hat{r} \in \mathcal{R}_{\hat{\mathfrak{B}}},$$

*where $\pi^*$ is an optimal policy in $\mathcal{M} \cup r$ and $\hat{\pi}^*$ is an optimal policy in $\widehat{\mathcal{M}} \cup \hat{r}$.*

The first condition states that for each reward in the exact feasible set, the best reward we could estimate in the recovered feasible set has a low error everywhere. This condition is a type of "recall": every possible true reward function needs to be captured by the recovered feasible set. The second condition ensures that there is a possible true reward function with a low error for every possible recovered reward function. This avoids an unnecessarily large recovered feasible set. This condition is a type of "precision": if we recover a reward function, it has to be close to a possible true reward function. Note, that Metelli et al. (2021) consider a similar optimality criterion in their Definition 5.1. However, they consider a known target environment; hence, our Definition 2 is a stronger requirement.

### 4.2 Feasible Rewards in Finite-horizon MDPs

Ng et al. (2000) characterize the feasible reward set implicitly in the infinite horizon setting, whereas Metelli et al. (2021) characterize it explicitly. Here, we provide similar results for a finite horizon.

**Lemma 3** (Feasible Reward Set Implicit). *A reward function $r$ is feasible if and only if for all $s, a, h$ it holds that: $A_{\mathcal{M} \cup r}^{\pi,h}(s,a) = 0$ if $\pi_h^E(a|s) \geq 0$ and $A_{\mathcal{M} \cup r}^{\pi,h}(s,a) \leq 0$ if $\pi_h^E(a|s) = 0$. Moreover, if the second inequality is strict, $\pi^E$ is uniquely optimal, i.e., $\Pi^*_{\mathcal{M} \cup r} = \{\pi^E\}$.*

**Lemma 4** (Feasible Reward Set Explicit). *A reward function $r$ is feasible if and only if there exists an $\{A_h \in \mathbb{R}_{\geq 0}^{\mathcal{S} \times \mathcal{A}}\}_{h \in [H]}$ and $\{V_h \in \mathbb{R}^{\mathcal{S}}\}_{h \in [H]}$ such that for all $s, a, h$ it holds that:*

$$r_h(s, a) = -A_h(s, a) \mathbb{1}_{\{\pi_h^E(a|s)=0\}} + V_h(s) + \sum_{s'} P(s'|s, a) V_{h+1}(s')$$

Here, the **first term** ensures there is an advantage function for $\pi^E$ and it is 0 for actions the expert takes and $A_h(s, a)$ for actions the expert does not take. The **second term** corresponds to reward-shaping by the value function.

### 4.3 Error Propagation

Next, we study the error propagation of estimating the transition model $P$ with $\widehat{P}$ and the expert policy $\pi^E$ with $\hat{\pi}^E$. In particular, we bound the estimation error on the reward as a function of the estimation errors of $\widehat{P}$ and $\hat{\pi}^E$, extending a result by Metelli et al. (2021) to the finite horizon setting.

**Theorem 5** (Error Propagation). *Let $(\mathcal{M}, \pi^E)$ and $(\widehat{\mathcal{M}}, \widehat{\pi}^E)$ be two IRL problems. Then, for any $r \in \mathcal{R}_{(\mathcal{M}, \pi^E)}$ there exists $\widehat{r} \in \widehat{\mathcal{R}}_{(\widehat{\mathcal{M}}, \widehat{\pi}^E)}$ such that:*

$$|r_h(s, a) - \widehat{r}_h(s, a)| \leq A_h(s, a)|\pi_h^E(a|s) - \widehat{\pi}_h^E(a|s)| + \sum_{s'} V_{h+1}(s')|P(s'|s, a) - \widehat{P}(s'|s, a)|$$

*and we can bound $V_h \leq (H - h)R_{\max}$ and $A_h \leq (H - h)R_{\max}$.*

In IRL, we cannot hope to recover the expert's reward function perfectly. Instead, we aim to estimate a reward function that leads to an optimal policy with performance close to the expert's policy under the (unknown) real reward function. For example, suppose a specific state $s$ is difficult to reach in the environment. In that case, the error on the reward function $r(s, \cdot)$ will not impact the performance of the induced policy much. Formally, we are interested in studying the error propagation to the optimal value function. The next lemma will be crucial for analyzing this.

**Lemma 6.** *Let $\mathcal{M}$ be an MDP\R, $r, \hat{r}$ two reward functions with optimal policies $\pi^*, \hat{\pi}^*$. Then,*

$$Q_{\mathcal{M} \cup r}^{\pi^*, h}(s, a) - Q_{\mathcal{M} \cup r}^{\hat{\pi}^*, h}(s, a) \leq \sum_{h'=h}^{H} \sum_{s', a'} \left( \eta_{\mathcal{M}, \pi^*}^{h, h'}(s', a'|s, a) - \eta_{\mathcal{M}, \hat{\pi}^*}^{h, h'}(s', a'|s, a) \right) (r_{h'}(s', a') - \hat{r}_{h'}(s', a'))$$

By combining this lemma with Theorem 5, we can decompose the error in the value function and Q-function into the error in estimating the transition model and the error in estimating the expert policy.

## 5 Recovering Feasible Rewards with a Generative Model

As a warmup, let us first study the sample complexity of a simple *uniform sampling* strategy with access to the generative model of $\mathcal{M}$. We assume we can query a generative model about a state-action pair $(s, a)$ to receive a next state $s' \sim P(\cdot|s, a)$ and an expert action $a_E \sim \pi^E(\cdot|s)$. This allows us to introduce key ideas and serves as a baseline to compare later results to. We adapt the infinite-horizon results by Metelli et al. (2021) to the finite-horizon setting, and our stronger PAC requirement in Definition 2. We first discuss how we can estimate the transition model and the policy (Section 5.1) before stating the sample complexity of the uniform sampling strategy (Section 5.2).

### 5.1 Estimating Transition Model and Expert Policy

In each iteration $k$, let $n_k^h(s, a, s')$ be the number of times we observed the transitions $(s, a, s)$ at time $h$ up to iteration $k$. Also, we define $n_k^h(s, a) = \sum_{s'} n_k^h(s, a, s')$, and $n_k^h(s) = \sum_a n_k^h(s, a)$. Then we can estimate the transition model and expert policy by

$$\widehat{P}_k(s'|s, a) = \frac{\sum_{h=1}^{H} n_k^h(s, a, s')}{\max(1, \sum_{h=1}^{H} n_k^h(s, a))} \qquad \hat{\pi}_{k, h}^E(a|s) = \frac{n_k^h(s, a)}{\max(1, n_k^h(s))}.$$

---

**Algorithm 1** AceIRL algorithm for IRL in an unknown environment.

---

1: **Input:** significance $\delta \in (0,1)$, target accuracy $\epsilon$, IRL algorithm $\mathscr{A}$, number of episodes $N_E$
2: Initialize $k \leftarrow 0, \;\; \epsilon_0 \leftarrow H/10$
3: **while** $\epsilon_k > \epsilon/4$ **do**
4:      Solve (convex) optimization problem (ACE) to obtain $\pi_k$
5:      Explore with policy $\pi_k$ for $N_E$ episodes, observing transitions and expert actions
6:      $k \leftarrow k+1$
7:      Update $\widehat{P}_k, \hat{\pi}_k, C_k^h$, and $\hat{r}_k \leftarrow \mathscr{A}(\mathcal{R}_{\hat{\mathfrak{B}}})$
8:      Update accuracy $\epsilon_k \leftarrow \max_a \hat{E}_k^0(s_0, a)$
9: **end while**
10: **return** Estimated reward function $\hat{r}_k$

---

In Appendix B.3 we derive Hoeffding's confidence intervals for the transition model and the expert policy. Combining these with Theorem 5, we can compute the uncertainty on the recovered reward as:

$$C_k^h(s,a) = (H-h)R_{\max} \min \left( 1, 2\sqrt{\frac{2\ell_k^h(s,a)}{n_k^h(s,a)}} \right),$$

where $\ell_k^h(s,a) = \log\left(24SAH(n_k^h(s,a))^2/\delta\right)$. We can show that for any pair of reward functions $r \in \mathcal{R}_{\mathfrak{B}}$ and $\hat{r} \in \mathcal{R}_{\hat{\mathfrak{B}}}$, the difference $|r_h(s,a) - \hat{r}_{k,h}(s,a)| \leq C_k^h(s,a)$. This uncertainty estimate will be a key component in all of our theoretical analysis.

### 5.2 Uniform Sampling Strategy

In each iteration $k$, the *uniform sampling* strategy allocates $n_{\max}$ samples uniformly over $[H] \times \mathcal{S} \times \mathcal{A}$. It estimates the reward uncertainty and stops as soon as $H \max_{h,s,a} C_k^h(s,a) \leq \epsilon$. The next theorem characterizes the sample complexity of uniform sampling with a generative model.

**Theorem 7** (Sample Complexity of Uniform Sampling IRL)**.** *The uniform sampling strategy fulfills Definition 2 with a number of samples upper bounded by:*

$$n \leq \tilde{\mathcal{O}}\left( H^5 R_{\max}^2 SA/\epsilon^2 \right),$$

*where $\mathcal{O}$ suppresses logarithmic terms.*

This sample complexity bound appears slightly worse than the one in Metelli et al. (2021), who find $(1-\gamma)^{-4}$ which would translate to $H^4$. This is, however, due to the fact that we consider reward functions that can depend on the timestep $h$. If we assume the reward function does not depend on $h$, we gain a factor of $H$, obtaining the same result as Metelli et al. (2021).

## 6 Active Exploration for Inverse Reinforcement Learning

Let us now turn to our original problem: recovering the expert's reward function in an unknown environment *without* a generative model. This problem is harder since we need to create an exploration strategy to acquire the desired information about the environment. We now propose a novel sample-efficient exploration algorithm for IRL that we call **Ac**tive **e**xploration for **I**nverse **R**einforcement **L**earning (AceIRL). The algorithm takes inspiration from recent works on reward-free exploration (Kaufmann et al., 2021) and exploration strategies in RL (Auer et al., 2008). We divide the explanation of the algorithm into two parts. First, we introduce a simplified version of the algorithm, which comes with a problem independent sample complexity result (Section 6.1). Next, we introduce the full algorithm, which considers the expected reduction of uncertainty in the next iteration to improve exploration and maintains a confidence set of plausibly optimal policies to focus on the most relevant regions (Section 6.2). The full algorithm provides a tighter, problem-dependent sample complexity bound (Section 6.3). Algorithm 1 contains pseudo-code of AceIRL, and Appendix B contains the detailed theoretical analysis including proofs of all results discussed here.

## 6.1 Uncertainty-based Exploration for IRL

The first idea of AceIRL is similar to reward-free UCRL (Kaufmann et al., 2021). Our goal is to fulfill the PAC requirement in Definition 2. Hence, we start from an upper bound on the estimation error between the performance of the optimal policy $\hat{\pi}^*$ for a reward $\hat{r} \in \mathcal{R}_{\hat{\mathfrak{B}}}$ in the recovered feasible set and the optimal policy $\pi^*$ for a reward function $r \in \mathcal{R}_{\mathfrak{B}}$ in the true MDP $\mathcal{M}$. We will then use this upper bound to drive the exploration. For each timestep $h$ and iteration $k$, we define the error:

$$\hat{e}_k^h(s, a; \pi^*, \hat{\pi}^*) = \left| Q_{\mathcal{M} \cup r}^{\pi^*, h}(s, a) - Q_{\mathcal{M} \cup r}^{\hat{\pi}^*, h}(s, a) \right|. \tag{1}$$

We can define an upper bound on these errors recursively with $C_k^H(s, a) = 0$ and

$$E_k^h(s, a) = \min\Big( (H - h)R_{\max}, C_k^h(s, a) + \sum_{s'} \widehat{P}(s'|s, a) \max_{a' \in \mathcal{A}} E_k^{h+1}(s', a') \Big). \tag{EB1}$$

It is straightforward to show that $\hat{e}_k^h(s, a; \pi^*, \hat{\pi}^*) \leq E_k^h(s, a)$ for any two policies $\pi^*, \hat{\pi}^*$. Using this error bound, we can introduce a simplified version of AceIRL that explores greedily with respect to $E_k^h(s, a)$. We call this algorithm "AceIRL Greedy". Note that this is equivalent to solving the RL problem defined by $\mathcal{M} \cup C_k^h$; hence, we can use any RL solver to find the exploration policy in practice. If we explore with this greedy policy, we can stop if:

$$4 \max_a E_k^0(s_0, a) \leq \epsilon. \tag{SP1}$$

We can show that when this stopping condition holds, the solution fulfills the PAC requirement 2. Furthermore, we show in Appendix B.4 that AceIRL Greedy achieves a sample complexity on order $\tilde{\mathcal{O}}\left(H^5 R_{\max}^2 SA/\epsilon^2\right)$, which matches the sample complexity of uniform sampling *with a generative model*. This is already a strong result implying that we do not need a generative model to achieve a good sample complexity for IRL. However, it turns out we can improve the algorithm further.

## 6.2 Problem Dependent Exploration

AceIRL Greedy is limited in two ways: (i) it explores states that have high uncertainty so far, whereas our goal is to reduce uncertainty *in the next iteration*, and (ii) it explores to reduce the uncertainty about all policies, whereas our goal is to reduce the uncertainty primarily about *plausibly optimal* policies. To address these limitations, we propose two modifications that yield the full AceIRL algorithm.

**Reducing future uncertainty.** The greedy policy w.r.t. $E_k^h$ explores states in which the estimation error on the Q-functions is large. However, note that this is not exactly what we want, namely, to reduce the uncertainty the most. In particular, if we explore for more than one episode before updating the exploration policy, we should choose an exploration policy that considers how the uncertainty will reduce during exploration. Ideally, we would explore with a policy that minimizes $E_{k+1}^h$. However, we cannot compute this quantity exactly. Instead, we can approximate it using our current estimate of the transition model. Concretely, if we have an exploration policy $\pi$, we can estimate the reward uncertainty at the next iteration as:

$$\hat{C}_{k+1}^h(s, a) = (H - h)R_{\max} \min\left(1, 2\sqrt{\frac{2\ell_k^h(s, a)}{n_k^h(s, a) + \hat{n}_\pi^h(s, a)}}\right),$$

where $\hat{n}_\pi^h(s, a) = N_E \cdot \eta_{\mathcal{M}, \pi}^{0, h}(s, a|s_0)$ is the expected number of times $\pi$ visits $(s, a)$ at time $h$ and $N_E$ is the number of episodes we will explore with $\pi$. We can use this estimate to find a policy that minimizes our estimate of $E_k^{h+1}$. While our original approach was akin to "uncertainty sampling", we now have a better way to measure the "informativeness" of choosing an exploration policy. This is a common pattern when designing query strategies in active learning (Settles, 2012). Note, that this argument does not rely on the IRL problem and can be used to independently improve algorithms for reward-free exploration (cf. Appendix D).

**Focusing on plausibly optimal policies.** By exploring greedily w.r.t. $E_k^h$, we reduce the estimation error of all policies. However, we are primarily interested in estimating the distance between policies $\pi^* \in \Pi_{\mathcal{M} \cup r}^*$ and $\hat{\pi}^* \in \Pi_{\mathcal{M} \cup \hat{r}}^*$ with $r \in \mathcal{R}_{\mathfrak{B}}$ and $\hat{r} \in \mathcal{R}_{\hat{\mathfrak{B}}}$. Of course, we do not know these sets, so we cannot use them directly to target the exploration. Instead, assume we can construct a set of

plausibly optimal policies $\hat{\Pi}_k$ that contains all $\pi^*$ and $\hat{\pi}_k^*$ with high probability. Then, we can redefine our upper bounds recursively as $\hat{E}_k^H(s, a) = 0$ and:

$$\hat{E}_k^h(s, a) = \min\left((H - h)R_{\max}, C_k^h(s, a) + \sum_{s'} \widehat{P}(s'|s, a) \max_{\pi \in \hat{\Pi}_{k-1}} \pi(a'|s')\hat{E}_k^{h+1}(s', a')\right), \text{ (EB2)}$$

In contrast to (EB1), we maximize over policies in $\hat{\Pi}_k$ rather than all actions. Acting greedily with respect to $\hat{E}_k^h(s, a)$ is equivalent to finding the optimal policy $\pi_k \in \hat{\Pi}_k$ for the RL problem defined by $\mathcal{M} \cup C_k^h$. To construct the set of plausibly optimal policies, we use an arbitrary IRL algorithm $\mathscr{A}$. We only assume that $\mathscr{A}$ will return a reward function $\hat{r}_k \in \mathcal{R}_{\hat{\mathfrak{B}}}$. Then, we can construct a set of plausibly optimal policies as $\hat{\Pi}_k = \{\pi | V_{\widehat{\mathcal{M}} \cup \hat{r}_k}^{*,}(s_0) - V_{\widehat{\mathcal{M}} \cup \hat{r}_k}^{\pi,}(s_0) \le 10\epsilon_k\}$. We show in Appendix B.5 that $\hat{\Pi}_k$ contains both $\pi^*$ and $\hat{\pi}_k^*$ with high probability. This choice is based on ideas by Zanette et al. (2019).

We can define a stopping condition analogously to (SP1):

$$4 \max_a \hat{E}_k^0(s_0, a) \le \epsilon. \tag{SP2}$$

Again, we can prove that if the algorithms stops due to (SP2), then $\mathcal{R}_{\hat{\mathfrak{B}}}$ respects Definition 2.

**Implementing AceIRL.** To implement the full algorithm, we need to solve an optimization problem:

$$\pi_k \in \operatorname*{argmin}_\pi \max_{\hat{\pi} \in \hat{\Pi}_{k-1}} \hat{E}_{k+1}^0(s_0, \hat{\pi}(s_0)) \tag{ACE}$$

The solution to this problem is the exploration policy that minimizes the uncertainty at the next iteration about plausibly optimal policies. This combines both modifications we just discussed. This problem might seem difficult to solve at first, but, perhaps surprisingly, it can be formulated as a convex optimization problem solvable with standard techniques (cf. Appendix B.6).

## 6.3 Sample Complexity of AceIRL

In this section, we present our main result about the sample complexity of AceIRL. The result is problem-dependent, and, in particular, depends on the advantage function $A_{\mathcal{M} \cup r}^{*,h}(s, a)$, where $r$ is the reward function in the exact feasible set $\mathcal{R}_{\mathfrak{B}}$ closest to the reward function $\hat{r}_k$ which belongs to the estimated feasible set $\mathcal{R}_{\hat{\mathfrak{B}}}$. The advantage function $A_{\mathcal{M} \cup r}^{*,h}(s, a)$ acts similarly to a suboptimality gap: the closer the value of the second best action is to the best action, the harder it is to identify the best action and infer the correct reward function.

**Theorem 8.** *[AceIRL Sample Complexity] AceIRL returns a ($\epsilon$, $\delta$, $n$)-correct solution with*

$$n \le \tilde{\mathcal{O}}\left(\min\left[\frac{H^5 R_{\max}^2 SA}{\epsilon^2}, \frac{H^4 R_{\max}^2 SA \epsilon_{\tau-1}^2}{\min_{s,a,h}(A_{\mathcal{M} \cup r}^{*,h}(s, a))^2 \epsilon^2}\right]\right)$$

*where $\epsilon_{\tau-1}$ depends on the choice of $N_E$, the number of episodes of exploration in each iteration. $A_{\mathcal{M} \cup r}^{*,h}(s, a)$ is the advantage function of $r \in \operatorname{argmin}_{r \in \mathcal{R}_{\mathfrak{B}}} \max_{h,s,a}(r_h(s, a) - \hat{r}_{k,h}(s, a))$, the reward function from the feasible set $\mathcal{R}_{\mathfrak{B}}$ closest to the estimated reward function $\hat{r}_k$.*

This result is the minimum of two terms. The first term is problem independent and it is achieved both by AceIRL Greedy and the full AceIRL. This bound matches the bound we saw previously with a generative model. Hence, AceIRL achieves the same results without access to the generative model. Using (ACE) can yield a better sample complexity, represented by the second term in the minimum. This bound depends on two main components: the ratio $\epsilon_{\tau-1}/\epsilon$ and the advantage function $A_{\mathcal{M} \cup r}^{*,h}(s, a)$. The ratio depends on the choice of $N_E$, the number of exploration episodes per iteration. If $N_E$ is small, then the $\epsilon$-ratio will be also small. If $N_E$ is large the algorithm will perform similarly to a uniform sampling strategy. Appendix B.5 provides the full proof of this theorem.

| | Uniform sampling (gener. model) | TRAVEL (gener. model) (Metelli et al., 2021) | Random Exploration | AceIRL Greedy | AceIRL (Full) |
|---|---|---|---|---|---|
| Four Paths (Figure 1) | $1900 \pm 71$ | | $17840 \pm 1886$ | | |
| $- N_E = 50$ | | $1560 \pm 76$ | | $24180 \pm 1747$ | $\mathbf{10780 \pm 1369}$ |
| $- N_E = 100$ | | $2000 \pm 0$ | | $32760 \pm 2172$ | $14080 \pm 1603$ |
| $- N_E = 200$ | | $4000 \pm 0$ | | $52000 \pm 4057$ | $16160 \pm 2033$ |
| Double Chain (Kaufmann et al., 2021) | $1980 \pm 66$ | | $23640 \pm 2195$ | | |
| $- N_E = 50$ | | $1120 \pm 46$ | | $16240 \pm 842$ | $\mathbf{11580 \pm 870}$ |
| $- N_E = 100$ | | $2000 \pm 0$ | | $22200 \pm 1329$ | $15440 \pm 1031$ |
| $- N_E = 200$ | | $4000 \pm 0$ | | $37200 \pm 1664$ | $20400 \pm 1629$ |
| Metelli et al. (2021): | | | | | |
| Random MDPs ($N_E = 1$) | $22 \pm 1$ | $27 \pm 1$ | $\mathbf{22 \pm 1}$ | $23 \pm 1$ | $\mathbf{21 \pm 1}$ |
| Chain ($N_E = 1$) | $78 \pm 2$ | $76 \pm 4$ | $161 \pm 8$ | $153 \pm 8$ | $\mathbf{142 \pm 9}$ |
| Gridworld ($N_E = 1$) | $43 \pm 2$ | $35 \pm 2$ | $\mathbf{45 \pm 2}$ | $\mathbf{46 \pm 3}$ | $48 \pm 2$ |

Table 1: Sample complexity of AceIRL compared to random exploration and methods that use a generative model. We show the number of samples necessary to find a policy with normalized regret less than $0.4$. We report means and standard errors computed over $50$ random seeds each. For each environment, we highlight in **bold** the method that achieves the best performance without access to a generative model. If multiple methods are within one standard error distance, we highlight all of them. Overall, AceIRL is the most sample efficient method without a generative model if $N_E$ is chosen small enough. In Appendix C.3, we show learning curves for all individual experiments.

# 7 Experiments

We perform a series of simulation experiments to evaluate AceIRL. We simulate a (deterministic) expert policy using an underlying true reward function, and compare it to the recovered reward functions. We provide a code to reproduce all of our experiments at https://github.com/lasgroup/aceirl.

Our main evaluation metric is a *normalized regret*:

$$\left(V^{\pi^*,0}_{\mathcal{M} \cup r}(s_0) - V^{\hat{\pi}^*,0}_{\mathcal{M} \cup r}(s_0)\right) / \left(V^{\pi^*,0}_{\mathcal{M} \cup r}(s_0) - V^{\bar{\pi}^*,0}_{\mathcal{M} \cup r}(s_0)\right),$$

where $\pi^*$ is the optimal policy for $\mathcal{M} \cup r$, $\hat{\pi}^*$ is the optimal policy for $\widehat{\mathcal{M}} \cup \hat{r}$, and $\bar{\pi}^*$ is the worst possible policy for $r$, i.e., the optimal policy for $\mathcal{M} \cup (-r)$.

We introduce the *Four Paths* environment shown in Figure 1, which consists of four chains of states that have different randomly sampled transition probabilities. One path has a goal with reward 1; all other rewards are 0. We also evaluate on *Random MDPs* with uniformly sampled transition dynamics and re-

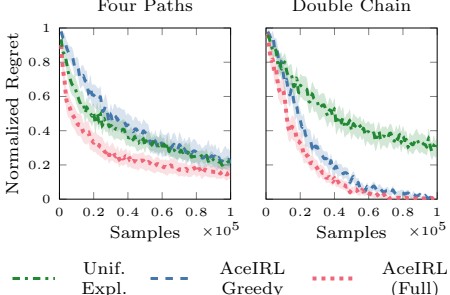

Figure 2: Normalized regret (lower is better) of the policy optimizing for the inferred reward in the estimated MDP as a function of the number of samples. The plots show the mean and $95\%$ confidence intervals computed using 50 random seeds. We use $N_E = 50$.

ward functions, the *Double Chain* environment proposed by Kaufmann et al. (2021), and the *Chain* and *Gridworld* environments proposed by Metelli et al. (2021). Appendix C.1 provides details on the transition dynamics and rewards of all environments.

We compare AceIRL and AceIRL Greedy to a uniformly random exploration policy, as a naive exploration strategy. Further, we consider uniform sampling with a generative model as well as TRAVEL (Metelli et al., 2021), which can be more sample efficient because they do not need to explore the environment. Note that TRAVEL is designed to learn a reward to be transferred to a known target environment. Instead, we use a modified version that uses the estimated MDP as a target. Appendix C.2 provides more details on our implementations, and we provide source code in the supplemental material.

Table 1 shows the sample efficiency of all algorithms in all environments, measured as the number of samples needed to achieve a regret threshold of $0.4$ (different thresholds yield similar conclusions; cf. Appendix C). AceIRL is the most sample efficient exploration strategy without access to a generative model; but the relative differences between the methods depend on the environment. In some cases, AceIRL even performs comparably to methods using a generative model, as the theory predicts.

In the *Four Paths* and *Double Chain* environments, we also vary the $N_E$ parameter. AceIRL performs better for small values at the computational cost of updating the exploration policy more often. If

$N_E$ is too large, using AceIRL can be as bad as a uniformly random exploration policy. Increasing $N_E$ hurts the performance of AceIRL Greedy more severely, which does not consider $N_E$ explicitly. Figure 2 shows the normalized regret as a function of the number of samples in *Four Paths* and *Double Chain*. In both cases AceIRL performs best. However, AceIRL Greedy is worse than random exploration in the *Four Paths* environment. Hence, we find that the problem dependent exploration strategy of the full algorithm significantly improves the sample efficiency.

## 8 Conclusion

We considered active inverse reinforcement learning (IRL) with unknown transition dynamics and expert policy and introduced AceIRL, an efficient exploration strategy to learn about both the dynamic and the expert policy with the goal of inferring the reward function as efficiently as possible.

Our approach is a crucial step towards IRL algorithms with theoretical guarantees, but future work is needed to move to more practical settings. In particular, it would be interesting to extend the approach to continuous state and action spaces (e.g. using function approximation), and to obtain an efficient algorithm that does not require solving convex optimization problems. From a theoretical perspective, it would be useful to derive a lower bound on the sample complexity of the IRL problem, to understand if the IRL problem is inherently more difficult than usual RL. Beyond IRL, we believe that some of our methods could be useful for other settings, such as reward-free exploration (cf. Appendix D).

Sample efficient IRL is a promising way to apply RL in situations where there is no well-specified reward function available. Of course, even robust IRL algorithms pose a risk of misuse. But, we are optimistic that these methods will overall lead to safer RL systems that can be used in real applications.

## Acknowledgements

This project has received funding from the Microsoft Swiss Joint Research Center (Swiss JRC), Google Brain, and from the European Research Council (ERC) under the European Union's Horizon 2020 research and innovation programme grant agreement No 815943. We thank Bhavya Sukhija for valuable feedback on an earlier draft.

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
