# OpenReview forum: "Active Exploration for Inverse Reinforcement Learning"
_NeurIPS.cc/2022/Conference — NeurIPS 2022 Accept_

### Official Review · Reviewer_Jvjh · 2022-06-29

**Rating:** 7
**Confidence:** 3
**Soundness:** 3 good
**Presentation:** 3 good
**Contribution:** 2 fair

**Summary:**

this paper proposes AceIRL, which is an IRL algorithm with active explorations in unknown environments. Authors give some novel analysis on this newly proposed method and empirically, AceIRL outperfroms some naive exploration methods.

**Questions:**

why not conduct experiments on some complex environment like MuJoCo?

**Strengths And Weaknesses:**

Strengths:

novle algorithm to explore the environment and the expert policy to infer a good reward signal.

provides two exploration strategies for AceIRL, and the algorithms are sound


Weaknesses:

the experiments seem not enough for evaluation. why not conduct experiments on some complex environment like MuJoCo?

---

> ### Author Response · Authors · 2022-08-01
> **Response to Jvjh**
>
> We thank the reviewer for the comments. We answer to the “Weakness” below.
>
> Weaknesses
>
> We agree that our experiments are conducted in simple domains and that experiments in more complex domains would be interesting. However, since the main goal of the paper is to highlight the theoretical aspects of the IRL setting, the experiments are intended to verify our theoretical findings. In particular, we wanted to show: (1) the AceIRL algorithm performs well also without access to a generative model, (2) the proposed exploration strategy outperforms the “random” exploration strategy (3) the sensibility of the N_E parameter. The extension towards more practical algorithms is an important future research direction that might require reconsidering/simplifying the optimization approach and function approximation techniques.

---

### Official Review · Reviewer_LBnb · 2022-07-10

**Rating:** 7
**Confidence:** 3
**Soundness:** 4 excellent
**Presentation:** 3 good
**Contribution:** 3 good

**Summary:**

The idea is to drive exploration to build the ideal set of demonstrations so that any IRL algorithm can reconstruct a reward function that leads to a policy as close as the optimal one, with a fixed limited number of examples.
The authors first set up a PAC criteria that their algorithm must verify in order to converge.
Then, they analyse three strategies
- A (oracle) uniform sampling as baseline
- A greedy exploration that picks transitions that reduce the IMMEDIATE uncertainty about the value associated with ANY policy.
- A smarter exploration that picks transitions that reduce the FUTURE uncertainty about the value associated with THE OPTIMAL policy.

They compare theoretical complexities and empirical performances of these different strategies.

**Questions:**

Cf minor critics above.

**Limitations:**

Not applicable.

**Strengths And Weaknesses:**

The paper is nicely written, the methodology is well driven and brings a solid solution to the problem.
I did not go through the proof, but the mathematical claims look intuitively correct.
The empirical results are reflecting the theoretical findings, and they are verified over 50 seeds.

I would only have minor critics:

1) The abbreviation PAC (Probability Approximately Correct) is never written without abbreviation (so I had to check the meaning in another paper).
2) In Theo. 1, A and V  are associated with which MDP’s reward? (the ground truth or the approximation?)
3) Detail: in lemma 3, “theorem 4” should be “theorem 1”.
4) L209: if you use the word “straightforward”, you have to at least explain why it is straightforward (in this case, triangular inequality with bellman equations?)
5) L256 Why is it “surprising”?

---

> ### Author Response · Authors · 2022-08-01
> **Response to LBnb**
>
> We would like to thank the reviewer for the positive feedback and for the comments. We answer the minor critics below.
>
> 1. Thanks for pointing it out. We will properly introduce PAC in the revised paper.
>
> 2. A and V are associated with the true reward function.
>
> 3. Thanks, we will fix this typo.
>
> 4. Correct, we will add the explanation in the revised paper.
>
> 5. We added “surprising” because it is not immediately obvious that the two-level optimization problem is a convex problem and one might expect the problem to be hard to solve.

---

### Official Review · Reviewer_uWZB · 2022-07-10

**Rating:** 7
**Confidence:** 4
**Soundness:** 3 good
**Presentation:** 4 excellent
**Contribution:** 3 good

**Summary:**

This paper considers the problem of active learning in explorations for inverse RL. And this paper provides necessary and sufficient conditions for solving such active learning problem.  They theoretical analyze the estimation error of the reward function caused by the error in transition function and expert policy. A novel algorithm AceIRL is proposed in this paper whose idea is to actively explore sample with large uncertainties for learning a better reward function. Some empirical experiment results are presented to support the theories. And the proposed method performs better than naive exploration strategies.

**Questions:**

I have a concern regarding the scalability of the proposed algorithm to large state and action spaces MDPs. The proposed transition model and expert policy model are calculated by using the frequency which is actually not sample efficient. And it may lead to a large variance estimation although they are unbiased. Can you provide some explanation for this problem?

And the idea of focusing on plausibly optimal policies is a good way to reduce sample complexity. But in the real experiments, there is no description for how you choose the optimal policy set. It will be great if you can present some details for your experiment.

In the experiments, only sample complexity is considered. It will be good if you can include the comparison of estimation errors.


**Limitations:**

Yes. The authors provides adequate information for this.

**Strengths And Weaknesses:**

Strengths: Overall, this paper is well written. The theorems are presented in a clear way. Sufficient number of theoretical analysis are provided to support the main result. I like the idea of applying a similar technique in UCRL to the inverse RL problem for estimating a better reward function. The proposed algorithm makes a lot of sense to me. The problem considered in this paper is promising since it is important to select samples in a efficient way for estimating the reward function. When the reward function is hard to learn, the proposed method has significant advantage over other random sampling based algorithm.
Weaknesses: The empirical result in this paper is not sufficient to show the benefits of active learning for IRL problems.

---

> ### Author Response · Authors · 2022-08-01
> **Response to uWZB**
>
> We would like to thank the reviewer for the positive feedback and constructive comments. We answer the reviewer’s questions one by one below.
>
> 1. (Scalability) We agree with the reviewer that the experiments are conducted in simple domains and the algorithm does not scale easily to large state-action spaces. However, we would like to underline that this is the first sample-efficient algorithm for IRL. The extension towards more practical algorithms is an interesting future research direction that might require reconsidering/simplifying the optimization approach and use of function approximation techniques.
>
> 2. (Plausibly optimal policies) To determine the set of plausibly optimal policies, we literally use the definition in line 249. This definition becomes a set of linear constraints in the optimization problem in Appendix B.6. We will make this clearer in the revised paper.
>
> 3. (Estimation error) We thank the reviewer for pointing this out. We did not evaluate reward estimation errors because there is not a unique way to define them in IRL (since more than one reward function can explain the expert’s behavior). Because we do not recover a unique reward function, we have to define the optimality criterion in Definition 2. So, the normalized regret in Figure 2 is the closest thing to an estimation error that we can evaluate.

---

### Official Review · Reviewer_qWnj · 2022-07-11

**Rating:** 7
**Confidence:** 2
**Soundness:** 4 excellent
**Presentation:** 4 excellent
**Contribution:** 3 good

**Summary:**

The paper explores the problem of active IRL, where the goal is to explore the environment to infer a good reward function. The paper addresses the Active IRL problem without assuming access to the transition model of the environment or the expert's policy. The paper proposes the Active Exploration for IRL (AceIRL) method and analyzes two sampling strategies. Finally, the paper evaluates AceIRL in 4 environments comparing normalized regret and sample efficiency to baselines.


**Questions:**

* Why is the sample complexity of AceIRL Greedy higher empirically than Uniform Sampling (gener. model) in the Four Paths and Double Chain environments in Table 1 when the theoretical sample complexity bounds are the same? In Four Paths and Double Chain environments, the sample complexity of AceIRL greedy is over an order of magnitude higher than that of Uniform Sampling (gener. model).
* How does the Random Exploration baseline work? I do not see it described in the text.
* Why does AceIRL Greedy underperform Random Exploration in the Four Paths environment? This is surprising since AceIRL Greedy tries to minimize the reward estimation error.
* Why are the results on the environments from [1] so different from the first two in Table 1? The gap between AceIRL and baselines is much larger in the first two environments, and AceIRL greedy outperforms AceIRL Full in some of the settings from [1]. What is the reason for this difference?

[1] A. M. Metelli et al. Provably efficient learning of transferable rewards. In Proceedings of International Conference on Machine Learning (ICML), 2021.

**Limitations:**

The paper discusses that the method is limited to discrete state and action spaces, and future work will extend it to continuous states and actions. The paper also states the efficiency limitations of iteratively solving convex optimization problems in the algorithm. Potential negative social impacts are also discussed.

**Strengths And Weaknesses:**

Strengths:
* AceIRL tackles the important problem of Active IRL without any generative model. No assumption of a generative model is an important improvement over baselines like TRAVEL. In my understanding, AceIRL is the first method for Active IRL without a generative model.
* Two versions of AceIRL are studied and theoretically analyzed with sample complexity bounds.
* The paper characterizes the goals of Active IRL with Definitions 1 and 2. Furthermore, the paper studies the impact of using learned generative models for the expert and transition model on the recovered reward in the finite horizon setting.
* The paper is the first to study Active IRL in the context of finite-horizon, undiscounted MDPs.

Weaknesses:
* In the related work, I would like to see more comparisons to how active IRL relates to the standard IRL problem. For example, what is the relationship between Active IRL methods and other IRL methods such as Apprenticeship IRL [4] or MaxEnt IRL [3]? Furthermore, in contrast to the claim on line 63, many recent IRL algorithms that use function approximation do not assume the underlying transition model is known, such as the f-IRL method [2]. Are these methods not applicable to the active IRL setting?
* How does Theorem 1 differ from Theorem 3.1 of [1] since both characterize the impact of error propagation from learned generative models?

[1] A. M. Metelli et al. Provably efficient learning of transferable rewards. In Proceedings of International Conference on Machine Learning (ICML), 2021.
[2]  Ni, Tianwei, et al. "f-irl: Inverse reinforcement learning via state marginal matching." arXiv preprint arXiv:2011.04709 (2020).
[3] Ziebart, Brian D., et al. "Maximum entropy inverse reinforcement learning." Aaai. Vol. 8. 2008.
[4] Abbeel, Pieter, et al. "Apprenticeship learning via inverse reinforcement learning." Proceedings of the twenty-first international conference on Machine learning. 2004.

---

> ### Author Response · Authors · 2022-08-01
> **Response to qWnj**
>
> We would like to thank the reviewer for the comments and suggestions. We answer the reviewer’s questions one by one, and then briefly comment on the weaknesses described by the reviewer.
>
> Questions
>
> 1. The matching sample complexity bound describes a worst-case instance. Both AceIRL Greedy and Uniform Sampling with a generative model do not have instance-dependent bounds. In the worst case, we do find similar sample complexity empirically, for example in Random MDPs. However, in individual instances, the sample complexity can be quite different. In the Four Paths and Double Chain environments, the generative model helps a lot because the agent starts in the middle and needs to spend a lot of samples to reach the ends of the different paths without a generative model.
>
> 2. Random Exploration always selects an exploration policy that chooses actions uniformly at random. We will make sure to mention this in the revised paper explicitly.
>
> 3. The main reason is that we choose N_E > 1, i.e., we explore for more than 1 episode using the same exploration policy. AceIRL Greedy will go where there is a high reward estimation error, but does not adapt this estimate over say 50 exploration episodes, which causes it to actually perform worse than random exploration. So, AceIRL Greedy will for example go down one path where there is high uncertainty 50 times, while random exploration randomizes its actions and thereby reduces uncertainty by more over 50 exploration episodes. The full AceIRL algorithm fixes that by modeling future uncertainty.
>
> 4. Similar to point 3, this is mainly explained since in the environments by [1] we use N_E=1 because they are very small. This means AceIRL Greedy updates its exploration policy for every exploration episode and, therefore, performs much better. Double Chain and Four Paths are larger environments where we choose N_E > 1 and get these qualitatively different results.
>
>
> Weaknesses
>
> - We will enhance the discussion of the connection of Active IRL to the standard IRL setting in the paper. In brief, our setup is primarily about how the data for IRL is collected and how transitions and expert policy are estimated from samples. Essentially, any IRL algorithm, such as MaxEnt IRL, can be used as a subroutine of AceIRL. IRL algorithms that do not assume access to the transition dynamics explicitly, usually use them implicitly somehow, e.g., in f-IRL through the observed state density. AceIRL can still be used to collect observations for f-IRL, even though it does not explicitly use the estimated transition model.
>
> - Our Theorem 1 states a similar result to Metelli et al.’s Theorem 3.1., only in the undiscounted finite-horizon setting rather than the discounted infinite horizon setting. We will make this connection clearer in the paper.

---

> > ### Comment · Reviewer_qWnj · 2022-08-06
> > **Response to Authors**
> >
> > Thank you for the response and clarifications. The paper would benefit from including this discussion on how Active IRL fits with standard IRL. The author's responses have addressed both of my stated weaknesses therefore I raise my score.

---

### Meta-Review · Area_Chair_DZoB · 2022-08-23

**Recommendation:** Accept
**Confidence:** Certain

**Metareview:**

This submission is solid as a work that introduces a novel IRL approach and analyzes its theoretical underpinnings. Its evaluation is done on very much toy problems though, and the metareviewer suggests reinforcing it with more complicated benchmarks, e.g., low-dimensional MuJoCo-based ones used in D4RL, to demonstrate that it works well even in continuous-state/-action settings.

**Award:**

No

---

### Decision · Program_Chairs · 2022-09-14

Accept